# Advancements in Telomerase-Targeted Therapies for Glioblastoma: A Systematic Review

**DOI:** 10.3390/ijms25168700

**Published:** 2024-08-09

**Authors:** Giovanni Pennisi, Placido Bruzzaniti, Benedetta Burattini, Giacomo Piaser Guerrato, Giuseppe Maria Della Pepa, Carmelo Lucio Sturiale, Pierfrancesco Lapolla, Pietro Familiari, Biagia La Pira, Giancarlo D’Andrea, Alessandro Olivi, Quintino Giorgio D’Alessandris, Nicola Montano

**Affiliations:** 1Department of Neurosurgery, Fondazione Policlinico Universitario A. Gemelli IRCCS, 00168 Rome, Italy; giovannipennisi91@gmail.com (G.P.); benedetta.burattini@gmail.com (B.B.); giacomo.piaserguerrato@gmail.com (G.P.G.); giuseppemaria.dellapepa@policlinicogemelli.it (G.M.D.P.); carmelo.sturiale@policlinicogemelli.it (C.L.S.); alessandro.olivi@unicatt.it (A.O.); quintinogiorgio.dalessandris@policlinicogemelli.it (Q.G.D.); nicolamontanomd@yahoo.it (N.M.); 2Department of Neurosurgery, F. Spaziani Hospital, 03100 Frosinone, Italy; biagialapira@hotmail.it (B.L.P.); gdandrea2002@yahoo.it (G.D.); 3Department of Human Neurosciences, Division of Neurosurgery, Policlinico Umberto I University Hospital, Sapienza, University of Rome, 00157 Rome, Italy; pietro.familiari@uniroma1.it; 4Nuffield Department of Surgical Sciences, University of Oxford, Oxford OX1 2JD, UK; pierfrancesco.lapolla@nds.ox.ac.uk

**Keywords:** glioblastoma (GBM), telomerase-targeted therapies, multi-target inhibitors, hTERT inhibitors, brain tumor

## Abstract

Glioblastoma (GBM) is a primary CNS tumor that is highly lethal in adults and has limited treatment options. Despite advancements in understanding the GBM biology, the standard treatment for GBM has remained unchanged for more than a decade. Only 6.8% of patients survive beyond five years. Telomerase, particularly the hTERT promoter mutations present in up to 80% of GBM cases, represents a promising therapeutic target due to its role in sustaining telomere length and cancer cell proliferation. This review examines the biology of telomerase in GBM and explores potential telomerase-targeted therapies. We conducted a systematic review following the PRISMA-P guidelines in the MEDLINE/PubMed and Scopus databases, from January 1995 to April 2024. We searched for suitable articles by utilizing the terms “GBM”, “high-grade gliomas”, “hTERT” and “telomerase”. We incorporated studies addressing telomerase-targeted therapies into GBM studies, excluding non-English articles, reviews, and meta-analyses. We evaluated a total of 777 records and 46 full texts, including 36 studies in the final review. Several compounds aimed at inhibiting hTERT transcription demonstrated promising preclinical outcomes; however, they were unsuccessful in clinical trials owing to intricate regulatory pathways and inadequate pharmacokinetics. Direct hTERT inhibitors encountered numerous obstacles, including a prolonged latency for telomere shortening and the activation of the alternative lengthening of telomeres (ALT). The G-quadruplex DNA stabilizers appeared to be potential indirect inhibitors, but further clinical studies are required. Imetelstat, the only telomerase inhibitor that has undergone clinical trials, has demonstrated efficacy in various cancers, but its efficacy in GBM has been limited. Telomerase-targeted therapies in GBM is challenging due to complex hTERT regulation and inadequate inhibitor pharmacokinetics. Our study demonstrates that, despite promising preclinical results, no Telomerase inhibitors have been approved for GBM, and clinical trials have been largely unsuccessful. Future strategies may include Telomerase-based vaccines and multi-target inhibitors, which may provide more effective treatments when combined with a better understanding of telomere dynamics and tumor biology. These treatments have the potential to be integrated with existing ones and to improve the outcomes for patients with GBM.

## 1. Introduction

Glioblastoma (GBM) is the most frequent primary Central Nervous System (CNS) tumor in adults [1]. It is among the most lethal forms of cancer and poses a formidable obstacle in terms of treatment.

Despite recent advances in understanding its biology, the treatment options remain limited. Only 6.8% of patients survive beyond 5 years of initial diagnosis [1,2].

The most notable progress in treating GBM occurred over ten years ago, when Temozolomide (TMZ) was added to postoperative radiotherapy (RT). This augmentation in median survival from 12.1 to 14.6 months was particularly beneficial to patients with methylation at the O6-methylguanine-DNA methyltransferase (MGMT) promoter in cancer DNA [3]. 

Despite the significant efforts of scientific research, nowadays the current standard of care for GBM has remained unchanged, with the gold standard of care including a combination of surgery, RT, and TMZ chemotherapy [4]. Thus, new therapies and specific tumor targets are urgently needed. However, this is especially challenging for GBM tumors due to their high heterogeneity in histopathological, molecular, genetic, and epigenetic features [5,6].

Telomerase is a ribonucleoprotein complex composed of multiple subunits [7]. The core catalytic subunit is the telomerase reverse transcriptase (hTERT), which is closely associated with a non-coding template telomerase RNA (hTR). The hTERT-hTR complex is bound to other proteins, including histones and ribonucleoproteins [7,8]. 

The identification of mutations in the telomerase reverse transcriptase (hTERT) promoter in 2013 revealed the most prevalent oncogenic mutation in glioblastoma, affecting up to 80% of cases [9,10]. Unlike other genetic changes, hTERT promoter mutations were observed to be clonal events in most cases and remained consistent between the samples taken before and after treatment. These mutations activate telomerase and maintain telomere length, thereby granting cancer cells an indefinite replicative capacity. Given these findings, targeting telomerase emerges as a compelling therapeutic strategy. Developing a successful telomerase inhibitor has been challenging despite its potential benefits as a target.

Our article examines the results obtained from available clinical trials of treatment with Telomerase-targeted therapies in adults with GBM through a systematic review of the literature in English. 

## 2. Material and Methods

The protocol for the systematic review presented herein was drafted in accordance with the guidelines for Preferred Reporting Items for Systematic Reviews and Meta-Analyses Protocols (PRISMA-P), as illustrated in Figure 1. A literature search was initiated on PubMed/Medline and Scopus in February 2024, with the final search conducted on 30 April 2024. 

As a search item, we searched for the following terms: “GBM”, “Glioblastomas”, “high-grade gliomas”, “hTERT”, and “telomerase”. Two authors, G.P. and P.B., independently evaluated the abstracts for eligibility. Any discordance was solved by consensus with the senior author (N.M.). No restrictions on the date of publication were made.

We included all the papers that discuss the target therapies for forward telomerase in adults (age > 18 years) who were affected by GBM. They discuss the target therapies for forward telomerase. Zhang et al. [12] report that 100% (11 out of 11) of the adult GBM displayed moderate levels of hTERT expression (value of 15.7), whereas 15% (4 out of 26) of the pediatric samples displayed weak to no expression (values of 20.0 to 24.0). We reviewed both in vivo and in vitro studies. Data regarding TERTp mutations in pediatric glioblastoma are lacking. TERTp mutations were reported to occur at a significantly lower rate in pediatric glioblastoma, ranging from 3 to 11%. This suggests that infinite proliferation of cancer cells is not achieved by TERTp mutation-mediated activation of telomerase [13]. 

We excluded studies published in different languages than English, review studies, and meta-analyses. We conducted a systematic abstract screening of the references in order to identify additional records. After eliminating the duplicate records, we conducted a thorough screening of a total of 777 records, and 46 full-text documents were assessed for eligibility. Ten of these were beyond the scope of the present review. During the final review process, we included a total of 36 papers.

## 3. Results

We assessed forty-six articles for eligibility, but ten of them were not relevant to this review. The pharmacological therapies discussed in these articles are summarized in Table 1, Table 2, Table 3, Table 4 and Table 5. In the following paragraphs, we categorized these therapies based on their primary mechanism of action.

### 3.1. Molecules That Inhibit hTERT Transcriptional Activity

Several in vitro and in vivo studies were conducted to decrease the activity of transcription of hTERT (Table 1, Figure 2).

**Table 1 ijms-25-08700-t001:** List of studies on pharmacological molecules with effect on hTERT inhibition at the transcriptional level.

Author	In Vitro	In Vivo	Drug	Mechanism of Action
Mirzazadeh et al. [14]	✔		Resveratrol (RSV)	Inhibition of hTERT transcription
Gurung et al. [15]	✔		Thymoquinone (TQ)	Inhibition of hTERT transcription
Khaw et al. [16]	✔		Curcumin	Inhibition of hTERT transcription
Khaw et al. [17]	✔		Plumbagin	Inhibition of hTERT transcription
Khaw et al. [18]	✔		Genistein	Inhibition of hTERT transcription
Khaw et al. [19]	✔		Trichostatin A (TSA)	Inhibition of hTERT transcription
Lin et al. [20]	✔	✔	Butylidenephthalide (BP)	Inhibition of hTERT transcription
Kiaris and Schally et al. [21]	✔	✔	MZ-5-156 (GH-RH antagonist)	Inhibition of hTERT transcription
Udroiu et al. [22]	✔		Epigallocatechingallate (EGCG)	Inhibition of hTERT transcription
Das et al. [23]	✔		Retinoids	Inhibition of hTERT transcription
Aquilanti et al. [5]	✔	✔	CRISPRi approach	Inhibition of hTERT transcription
Ergüven et al. [24]	✔	✔	Suramin	Inhibition of hTERT transcription

Khaw AK et al. conducted in vitro studies examining the effects of various biological molecules on GBM cell lines through telomerase modulation [16,17,18,19]. They tested curcumin, plumbagin, genistein, and trichostatin A (TSA), a specific histone deacetylase (HDAC) inhibitor, on both radioresistant (KNS60 and U251MG) and radiosensitive (GGM A172) GBM cell lines, as well as on medulloblastoma cells (ONS76). They used RT-PCR to investigate the expression of hTR and telomeric repeat amplification protocol (TRAP) to detect the TA. These compounds reduced hTERT mRNA levels via transcription inhibition. Except for TSA, where its HDAC inhibition capacity was mechanistically responsible for this effect, the mechanism of action has not been elucidated. The author hypothesized that the reduction in cell viability was due to cell-cycle arrest at the G2/M checkpoint.

In addition, phytochemical compounds were tested as an hTERT inhibition with interesting results [14,15]. In detail, Mirzazadeh et al. found that RSV significantly decreased hTERT mRNA expression in the GBM cell line (U87MG) compared to controls, leading to reduced cell viability [14]. The authors utilized RT-PCR to study the inhibition rate of RSV on hTERT gene expression. However, it is necessary to conduct Western blot analysis for translational evaluation, as well as a TRAP assay, to determine whether RSV decreases telomerase activity following the downregulation of the mRNA variant transcript.

Gurung RL et al. investigated the effects of thymoquinone (TQ), a monoterpene, on the GBM cell lines. They found that the TQ induced cell death in GBM cell lines by inhibiting hTERT and reducing telomerase activity (TRAP protocol) [15]. TQ was more effective in DNA-PKcs (DNA-dependent protein kinase) knockout lines (M059J) than in DNA-PKcs wild-type lines (M059K). This result highlights that the telomerase shortening is probably dependent on the status of DNA-PKcs. 

Udroiu et al. investigated the long-term effects of epigallocatechin gallate (EGCG) on radioresistant cells (U251MG). EGCG significantly reduced hTERT mRNA, telomerase activity (real-time quantitative-telomerase repeat amplification protocol assay), and cell growth rate, inducing senescence and telomere-independent genotoxicity [22].

Furthermore, the Butylidenephthalide (BP), the main component of the chloroform extract of Angelica sinensis was investigated by Lin et al. [20]. The authors studied the dose-dependent effects of BP on hTERT in glioma cells (DBTRG-05MG and GBM 8401). Their findings suggest that BP inhibits proliferation and induces senescence in human GBM by downregulating hTERT expression and consequently telomerase activity. They highlight that the treatment with BP with a specific concentration (25–100 µg/mL) reduced hTERT mRNA and telomerase activity within 48 h, respectively, as investigated using RT-PCR and the TRAP assay. This effect is modulated by downregulating Sp1 (binding site located in the hTERT core promoter region) and upregulating p16/p21 (cell-cycle regulatory proteins that are associated with senescence). Their hypothesis was supported by a mouse xenograft model, in which BP repressed telomerase and inhibited tumor proliferation, resulting in tumor senescence [20].

Kiaris and Schally conducted a study on the growth hormone-releasing factor (GH-RH) antagonist MZ-5-156 and its effects on glioma U87MG cells. The research found that the drug significantly reduced hTERT expression and telomerase activity, respectively, as investigated using RT-PCR and the TRAP assay both in vitro and in vivo (xenograft with athymic nude mice) [21]. However, the study did not demonstrate its effects on GBM cell tumorigenicity and proliferation, and the exact mechanism of action remains unclear.

Das et al. studied the effects of all-trans retinoic acid (ATRA) and 13-cis retinoic acid (13-CRA) on telomerase activity in C6 rat glioma cells (TRAP assay). The results demonstrated that the retinoids induced astrocytic differentiation with downregulation of telomerase activity and worked synergistically to enhance the sensitivity of cells to the cytotoxic agent IFN-gamma and the cytostatic agent (Taxol-TXL) for apoptosis [23]. 

Aquilanti et al. reported the application of CRISPRi (clustered regularly interspaced palindromic repeats interface) for the transcriptional silencing of TERT exon 1 and the TERT promoter in GBM cell lines (98G, LN18, and SF295) and patient-derived models [5]. They found that TERT promoter-mutant glioblastoma cells are dependent on telomerase and showed typical features of telomere crisis when telomerase is lost. They used a doxycycline-inducible CRISPR interference system to knock down *TERT* expression in vivo early and late in tumor development. Using orthotopic xenograft models, they also demonstrated that only animals with low tumor burden experience a survival advantage from telomerase inhibition. These findings support the importance of preclinical and eventually clinical research on anti-telomerase compounds for treating glioblastoma. They also aid in identifying the patient population that would benefit most from this treatment strategy. Controversial data were found by Mine Ergüven et al. about the application of Suramin to inhibit telomerase activity in a xenograft model (subcutaneous injection of C6 glioma cell line in rat Wistar) [24]. The authors found that Suramin inhibited telomerase activity (as investigated using the TRAP assay) in vitro cell lines in several tumor cell lines, except for brain tumors. In contrast, they reported that Suramin increases telomerase activity in a C6 glioma brain tumor. They reported this effect such as a hormetic effect on cancer cells grown. Examples of drugs that exhibit hormetic effects in vivo include resveratrol, suramin, and tamoxifen [25].

### 3.2. Molecules That Inhibit Direct or Indirect the hTERT

Several articles investigated the effects of molecules on the direct inhibition of hTERT [26,27,28]. On the other hand, other indirect mechanisms of inhibition of hTERT were reported [29,30,31]. Regarding the direct inhibition of hTERT, Lavanya et al. investigated the in vitro effects of BIBR1532, a specific telomerase inhibitor, on the GBM cell line LN18 [26] (Table 2 and Figure 3). 

**Table 2 ijms-25-08700-t002:** List of studies on pharmacological molecules with effect on hTERT with direct or indirect inhibition.

Author	In Vitro	In Vivo	Drug	Mechanism of Action
Lavanya et al. [26]	✔		BIBR1532	Direct Inhibition hTERT
Biray Avci et al. [28]	✔		BIBR1532	Direct Inhibition hTERT
Ahmad et al. [29]	✔	✔	Costunolide	Indirect hTERT Inhibition
Gurung et al. [30]	✔		MST-312	Indirect hTERT Inhibition
Takahashi et al. [32]	✔	✔	Eribulin	Indirect hTERT Inhibition
Cheng et al. [33]	✔		Arsenico	Indirect hTERT Inhibition

BIBR1532 is a non-peptidic, non-nucleoside small molecule telomerase inhibitor that specifically binds to the active site of hTERT. When administered across a wide dose range (25–200 μM), BIBR1532 demonstrated a dose-dependent reduction in cell viability and induced cytotoxicity. Using flowcytometry and the TRAP assay, they reported, respectively, the induction of apoptosis and a reduction in telomerase activity. This induction of apoptosis was associated with the downregulation of telomerase activity due to the post-translational modification of hTERT. 

Building on these findings, Biray Avci et al. investigated the epigenetic effects of BIBR1532 on the GBM cell line U87MG. In treated cells, compared with the control group, there was an increased expression of epigenetic regulatory proteins, such as histone deacetylases (HDACs) and DNA methyltransferases [28]. Flow cytometry (Annexin V-FITC kit) and total RNA isolation were performed to analyze the apoptosis index and gene expression. These results highlight that BIBR1532 is effective in altering the epigenetic mechanisms involved in telomerase expression in U87MG cells.

Another molecule with an inhibitor effect on hTERT is Costunolide (CS). Its telomerase inhibitor effects have been reported in several in vitro studies on breast cancer cells and other solid tumors [31,34]. Its inhibition mechanism is probably mediated by the upregulation of p21 and p53 and leads to the anti-proliferative effects of tumor cells [31]. 

Additionally, CS has been shown to induce cell apoptosis through both direct and indirect pathways. Regarding the application of CS in glioma studies, Ahmad et al. demonstrated that CS induces p53-mediated glioma cell death via reactive oxygen species (ROS) induction. They used glioma cell lines (A172, U87MG, and T98G) for in vitro studies (MTS assay, ROS assay, etc.) and xenograft with nude mice for in vivo studies. Specifically, p53-wildtype glioma cell lines were sensitive to the CS treatment, whereas the p53-mutant cell lines were not [29]. 

Gurung et al. investigated the in vitro effects of MST-312, a chemically modified derivative of green tea epigallocatechin gallate (EGCG), on telomerase activity. When medulloblastoma cells (ONS76) and GBM multiforme cell lines (M059K, KNS60) were treated with MST-312, the telomerase activity was reduced by approximately 50% (the TRAP assay). However, the levels of hTERT mRNA and protein remained unchanged. Interestingly, when MST-312 was withdrawn, there was a 95% recovery of basal telomerase activity within 72 h, indicating that MST-312 acts as a competitive telomerase inhibitor in brain tumor cells. Since MST-312 binds to telomere sites on DNA, it may trigger the DNA damage response pathway. Moreover, prolonged exposure to this telomerase inhibitor resulted in resistant cell subpopulations, suggesting that an effective strategy could involve combining telomerase inhibition with the DNA repair pathway blockade [30].

Takahashi et al. tested the effectiveness of Eribulin against GBM cells with TERT mutations. The Eribulin mesylate is a fully synthetic analog of halichondrin B, a natural product isolated from marine sponges and originally developed as a microtubule inhibitor [32]. Eribulin is currently approved in more than 60 countries for the treatment of refractory breast cancers and liposarcomas [35]. This molecule has been identified as a specific inhibitor of hTERT-RNA-dependent RNA polymerase (TERT-RdRP). The specific biological roles of hTERT-RdRP are not well understood, but they might involve maintaining heterochromatin, catabolizing mitochondrial ROS, or synthesizing siRNA [32,36]. 

Takahashi et al. demonstrated that Eribulin exhibited significant anticancer activity against GBM cells both in vitro and in vivo, leading to a significant extension of survival in mice with intracerebral GBM xenografts (nude mice). Eribulin effectively penetrated the brain tumor tissues and maintained high concentrations for over 24 h post-administration with uniform distribution. Beyond its role as a microtubule inhibitor, the inhibition of TERT-RdRP (measured with a specific immunoprecipitation–RdRP assay) may contribute to Eribulin’s potent anti-GBM effects [32]. These findings suggest that Eribulin could be a promising therapeutic option for GBM.

Ye Cheng et al. investigated the effectiveness of arsenic trioxide (As_2_O_3_) in some GBM cell lines (U87, U251, SHG44, and C6) by applying the TRAP assay, the flowcytometric assay, the immunofluorescence test, and the MTT assay [33]. Arsenic, a naturally occurring compound, has been used as a therapeutic agent since the 15th century and was found to be effective in treating acute promyelocytic leukemia in the 1970s [33,37]. The author discovered that As_2_O_3_ significantly increases cellular senescence in a dose-dependent manner. Various factors can induce cellular senescence, such as the suppression of telomerase, damage to telomeres, and chromosomal damage, with telomere dysfunction being the primary cause. Their observations showed notable increases in the proteins p53 and p21, which align with the presence of cellular senescence. As_2_O_3_ causes telomere dysfunction, leading to cell apoptosis, G2/M cell cycle arrest, and senescence through mechanisms involving p53 and p21.

### 3.3. Molecules Acting through hTR Inhibition

As previously described, the main catalytic subunit of telomerase reverse transcriptase (hTERT) is closely associated with a non-coding RNA component of telomerase (hTR). Several authors [38,39,40] have investigated the potential to interfere with hTR and consequently with cell proliferation. In this context, Imetelstat (IMT) is one of the most studied molecules for its promising results [38] (Figure 3 and Table 3). This compound is a short-chain oligonucleotide that binds with high affinity to the hTR of hTERT, leading to a dose-dependent and reversible inhibition of telomerase activity (TA). This investigation was conducted by Marian et al., and they reported a telomerase-specific inhibitor effect of IMT, as measured by the TRAP assay [38]. Although considered the most promising telomerase inhibitor, IMT has not yet received FDA approval. The author found that IMT had an impact on neurosphere cultures, reducing the TA by 50% after 72 h at a dose of 0.45 μM and completely at a dose of 4 μM. TA recovery occurred within 12 days after stopping the drug, indicating that the effects of IMT are reversible. Furthermore, with such promising data, it is important to note that a 72 h pre-treatment with IMT enhanced the antitumor effects of TMZ and radiation therapy in vitro. Following the orthotopic xenograft of the neurosphere, they administrated IMT at a dosage of 30 mg/kg intraperitoneally three times per week, resulting in a reduction in tumor activity by 70% within 3–5 days (in vivo test) [38]. 

Considering this result, Ferrandon et al. conducted an in vivo study to assess the therapeutic effectiveness of combining IMT and RT in a murine GBM orthotopic model (U87MG) [39]. They observed a significant decrease in tumor volume and TA (TRAP assay) after 28 days of treatment compared to the control groups. Additionally, they found a significant correlation between TA and the reduction in tumor volume. In a subsequent in vivo experiment comparing different treatment regimens (RT alone, IMT alone, and RT + IMT), the combination therapy was the most effective in extending animal survival [39]. Lastly, Ozawa et al. investigated the in vivo antitumor effect of the hTR inhibitor GRN163, which is an analog of IMT, in orthotopic xenografts of GBM cells (U251, U87) in nude rats [40]. In this study, the rats were treated with a 7-day infusion of either 150 or 500 nmol of GRN163, and they showed that both doses of GRN163 prolonged the survival of the rats as compared to the survival of the control group. The same compound was also tested by Hashizume et al. [41]. They demonstrated that GRN163 reduces the TA (TRAP assay) in vitro and the intranasal delivery of the GRN163 efficiently distributed into an intracerebral tumor and inhibited tumor growth in vivo. The experiment resulted in the prolonged survival of athymic rats without any apparent toxicity. 

**Table 3 ijms-25-08700-t003:** List of studies on pharmacological molecules with effect on hTR inhibition.

Author	In Vitro	In Vivo	Drug	Mechanism of Action
Marian et al. [38]	✔	✔	Imetelstat	hTR Inhibition
Ferrandon et al. [39]		✔	Imetelstat	hTR Inhibition
Ozawa et al. [40]		✔	GRN163	hTR Inhibition
Hashizume et al. [41]	✔	✔	GRN163	hTR Inhibition

### 3.4. Molecules That Inhibit Shelterin and/or Stabilize the G-Quadruplex Structure at the 3′ Telomere End

Mammalian telomeres consist of repeated TTAGGG sequences bound by the shelterin complex, including TRF1, TRF2, POT1, TPP1, and RAP1 [42] (Figure 3 and Table 4).

**Table 4 ijms-25-08700-t004:** List of studies on pharmacological molecules with effects on shelterin or the stabilization of the G-quadruplex structure at the 3′ telomere end.

Author	In Vitro	In Vivo	Drug	Mechanism of Action
Bejarano et al. [43]	✔	✔	TRF1 Inhibition	Sheltering proteins Inhibition
Zhou et al. [44]	✔		BRACO-19	G-quadruplex stabilization
Lagah et al. [45]	✔		RHPS4	G-quadruplex stabilization
Berardinelli et al. [46]	✔		RHPS4	G-quadruplex stabilization
Hasegawa et al. [47]	✔	✔	Telomestatin	G-quadruplex stabilization
Merle et al. [48]	✔		N-methylated triflate (TAC)	G-quadruplex stabilization

Research suggests that inhibiting TRF1 could be an alternative to telomerase inhibitors for targeting telomeres independently of their length. TRF1 directly binds to TTAGGG telomere repeats and is essential for telomere protection [49].

Bejarano et al. confirmed the antitumor effect of inhibiting TFR1 on the GBM cell line. They tested several TRF inhibitors (ETP-47228, ETP-47037, and ETP-50946). Inhibiting TRF1 showed enhanced antitumor effects when combined with gamma-irradiation and TMZ. However, the TRF1 inhibitors were found to be unable to cross the blood–brain barrier, limiting their potential clinical use [43].

The use of G-quadruplex (G4) ligands to target telomeres shows promise in cancer treatment. G4 ligands bind to DNA secondary structures called G-quadruplexes, which can be very stable under physiological conditions. These ligands interact with telomeres, induce telomere uncapping, and indirectly affect telomerase function. Some G4 ligands have been found to induce proliferation arrest and apoptosis in GBM cell lines [50,51] (Figure 4 and Table 4).

One of the G-quadruplex (G4) ligands tested in GBM cell lines is the BRACO-19, as reported by Zhou et al., who investigated the antitumor effects of this compound, which stabilizes the telomeric quadruplex DNA structure [44]. When the 3′-overhang of telomeric DNA forms a quadruplex structure, it cannot be extended by telomerase, so compounds that stabilize this structure inhibit telomerase activity. The study showed that U87, U251, SHG44, and C6 rat glioma cells exhibited a dose-dependent cytotoxic effect after 72 h of BRACO-19 treatment, while normal human astrocytes did not show decreased viability, indicating the selective killing of glioma cells. The cells treated with BRACO-19 displayed typical signs of anaphase bridges (measured with cytogenetic analysis and the telomere Tdt assay), suggesting telomere uncapping and the dissociation of telomere-binding proteins. These changes led to cell-cycle arrest in the G0–G1 phase and apoptotic death.

Additionally, the compound called Pentacyclic 3,11-difluoro-6,8,13-trimethyl-8H-quino[4,3,2-kl], a-cridinium methosulfate (RHPS4), was tested as a G4 ligand by several authors [45,46]. This compound binds and stabilizes telomeric DNA, leading to the blockage of replication forks at telomeres and consequently to telomere dysfunctionalities.

Lagah et al. reported an in vivo validation (medulloblastoma cells and U87 GBM) of the RHPS4. They reported a reduction in telomerase activity (measured with the TRAP assay), decreased cell viability (flow cytometric assay), and cell-cycle arrest [45]. 

Furthermore, this compound has been found to enhance the effects of radiation on GBM cells, making them more susceptible to treatment. Regarding the sensibilization effect, Berardinelli et al. investigated the combination of RHPS4 with carbon ion beam treatment, with interesting results. In detail, they found a significantly potentiated radiation effect in terms of cell killing, delayed rejoining of DNA double-strand breaks, chromosome aberrations, and G2/M-phase accumulation in GBM cells [46]. 

Merle et al. investigated the potential of N-methylated triflate (TAC), a novel G-quadruplex ligand, as a radiosensitizer for glioblastoma multiforme (GBM) under in vitro conditions. The study focused on two human GBM cell lines (SF763 and SF767), known for their resistance to radiation. The effects of TAC treatment on various cellular processes, such as telomere length, cell proliferation, apoptosis, cell cycle, gene expression, and DNA damage response were examined using the TRAP assay, flow cytometric Annexin Kit for apoptosis analysis, and RT-PCR. The results showed that low concentrations of TAC inhibited GBM cell proliferation and enhanced radiation-induced cell killing. The TAC treatment also led to DNA damage and altered the expression of telomere-related genes without significantly affecting telomere length. Additionally, the TAC treatment slowed down DNA repair processes and increased chromosomal aberrations when combined with radiation [48].

Hasegawa et al. proposed and explored the effectiveness of Telomestatin on glioma stem cells (GSCs) that drive tumor growth and recurrence [47]. Telomestatin, a natural compound, impairs GSC growth by disrupting transcription and triggering a DNA damage response. It stabilizes G-quadruplexes (G4), four-strand nucleic acid structures, both in vitro and in vivo. However, the reason for the selective DNA damage response in GSCs is unclear. Their study shows that GSCs are more vulnerable to Telomestatin-induced telomere dysfunction and replication stress compared to non-stem glioma cells (NSGCs). The telomerase inhibitor BIBR1532 did not selectively inhibit GSC growth, indicating that Telomestatin induces telomere dysfunction independently of telomerase [47]. 

### 3.5. Direct Modulation Therapies for hTERT Gene Expression

Several modalities were explored for gene expression modulation therapies able to reduce telomerase activity [26,52,53,54,55] (Table 5).

**Table 5 ijms-25-08700-t005:** List of studies on pharmacological molecules with direct modulation of hTERT gene expression.

Author	In Vitro	In Vivo	Gene Interference Strategy	Mechanism of Action through Gene Interfernce
Mancini et al. [55]	✔	✔	siRNA GABPβ1L inhibition	Inhibition of hTERT transcription
George et al. [52]	✔	✔	hTERT siRNA + IFN-γ	Indirect hTERT Inhibition
Falchetti et al. [53,54]	✔	✔	hTERT siRNA	Indirect hTERT Inhibition
Lavanya et al. [26]	✔		hTERT siRNA	Indirect hTERT Inhibition
Wang et al. [56]	✔	✔	Anti-miR21	Inhibition of hTERT transcription
Kim et al. [57]	✔		shMUC1	Indirect hTERT Inhibition
Vinchure et al. [58]	✔		miR-490	Sheltering proteins Inhibition

Mancini et al. focused on the genetic disruption of the GA-binding protein (GABP) subunit β, called GABPβ1L (β1L), which is essential for the normal development of proteins complexes, resulting in hTERT silencing. The researchers used GBM cell lines with mutant hTERT promoters (GBM1, T98G, LN229) and control lines with wild-type hTERT promoters (NHAPC5, HCT116, HEK293T) [55].

Disrupting β1L using siRNA and CRISPR-Cas9 to reduce hTERT expression leads to telomere loss and cell death, specifically in TERT promoter mutant cells (measured with a telomere qPCR protocol and flow cytometry [59]). In vivo experiments supported these findings, showing decreased tumor growth and prolonged survival in mice with xenografts. Although short-term cultures of all TERT-mutant cells showed reduced growth and viability, the long-term cultures of T98G and LN229 clones revealed surviving populations, indicating potential mechanisms for escaping β1L inhibition. In a study by Kim et al., the effects of suppressing mucin1 (MUC1) in GBM cells were investigated by Kim et al. [57]. MUC1 is a type I transmembrane protein that is overexpressed in most human epithelial cancers and is involved in epithelial–mesenchymal transition (EMT) and tumor progression. RNA sequencing of paired normal brain and tumor tissue from 30 patients revealed that MUC1 was significantly upregulated in glioma, regardless of WHO grade. GBM cells (U373, T98G, and A172) with reduced MUC1 (shMUC1) showed decreased cell proliferation and increased apoptosis (Annexin V Flow cytometric assay). Transcriptome analysis of naive and shMUC1 glioma cells indicated that MUC1 primarily regulates EMT and telomere-related pathways. In MUC1-knockout GBM cells, both hTERT expression and telomerase activity were reduced (measured with TRAP assay), while telomere restriction fragment (TRF) analysis showed slightly increased telomere length (RT-PCR and qPCR). This knockdown induced the alternative lengthening of the telomere (ALT) pathway as an escape mechanism, characterized by the presence of extrachromosomal telomeric circular DNA. These findings suggest that MUC1 depletion in GBM cells shifts the telomere maintenance mechanism from classic telomerase activation to the ALT pathway [57].

The application of RNA interference has been widely investigated in the past decades by several authors [53,54,60]. Using this technique, successful gene silencing can be achieved either through the introduction of synthetic, small interfering RNA (siRNA) oligonucleotides or their expression through a plasmid vector carrying a specific siRNA cDNA [52]. 

Regarding this technique, George et al. explored the potential combination of the recombinant plasmid-carrying hTERT siRNA during IFN-γ treatment in glioma cell lines (SNB19 and LN18) [52]. This approach was justified by the knowledge of the interaction between IFN-γ and hTERT. The researchers injected a recombinant plasmid carrying hTERT siRNA during IFN-γ treatment in glioma cell lines (SNB19 and LN18). In the co-treated cells, the levels of hTERT mRNA and protein were reduced, and angiogenesis ability was impaired. This was evidenced by the inhibition of capillary-like network formation when the cells were co-cultured with endothelial cells in vitro. In vivo studies confirmed that IFN-γ-treated cells with hTERT inhibition showed reduced angiogenesis and tumorigenic ability.

An Italian research group studied the role of telomerase in GBM aggressiveness by inhibiting hTERT using hTERT short interfering RNAs (si1-hTERT, si2-hTERT) in two GBM cell lines (TB10 and U87MG) [53,54]. 

The results of the experiments showed that the treated GBM cells had reduced hTERT mRNA, telomerase activity, and telomere length when tested in vitro using RT-PCR, TRAP, and telomere restriction fragment (TRF) assays. However, the cell growth rates were similar between the treated and control groups. In vivo experiments using subcutaneous and orthotopic xenografts indicated significantly impaired tumor growth in the si-hTERT TB10 and U87 GBM cells. Additionally, a significant reduction in angiogenesis was observed in the treatment group in the orthotopic xenografts. The role of telomerase in GBM angiogenesis was confirmed in co-xenografts of human umbilical vascular endothelial cells and TB10 GBM cells. The endothelial cells survived only when telomerase was upregulated and in the presence of tumor cells, while telomerase inhibition significantly reduced their survival.

Similarly, Lavanya et al. used the siRNA in an in vitro experiment [26]. They downregulated hTERT by siRNA in the LN18 glioma cell line. The inhibition of hTERT was effective in significantly decreasing cell viability 24 h post-transfection. Additionally, annexin V/PI double staining and Fluorescence-Activated Cell Sorting (FACS) analysis showed a notable increase in apoptosis levels. Wang et al. investigated the application of Anti-microRNAs on the modulation of hTERT expression [56]. MicroRNAs (miRNAs), which are approximately 20–22 nucleotides in length, are small, highly conserved noncoding RNA molecules. They regulate protein expression by cleaving or repressing the translation of target mRNA. Growing studies have indicated that miRNAs could function as oncogenic miRNAs, with miR-21 being overexpressed in various cancers, such as breast, lung, colon, and glioblastoma (GBM) [56,58,61].

Anti-microRNA-21 (as-miR-21) was used by Wang et al. to demonstrate miR-21 positive regulation over hTERT in GBM cell lines (U87 and LN229) [56]. The MTT assay, cell cycle analysis, and apoptosis analysis (using the flow cytometric Annexin V assay) demonstrated that the decrease in miR-21 levels led to inhibited cell growth in both U87 and LN229 GBM cells. Additionally, the reduction in miR-21 resulted in decreased expression of human telomerase reverse transcriptase (hTERT), as well as suppressed STAT3 expression and STAT3 phosphorylation. Finally, the knockdown of miR-21 considerably inhibited tumor growth and diminished the expression of STAT3 and hTERT in the xenograft model.

Vinchure et al. examined the impact of another microRNA on hTERT gene expression, specifically miR-490. Their findings indicated that miR-490 is a miRNA that is suppressed through epigenetic mechanisms in GBM cell lines (U87MG and T98G). They also discovered that miR-490 acts as a tumor-suppressor miRNA by inhibiting the TGF-β pathway-mediated epithelial-to-mesenchymal transition (EMT). In this context, they found that miR-490 controls telomere maintenance (TMM) in GBM cells. Moreover, miR-490 targets several genes (TRF2 of the shelterin complex, TNKS2, and SMG1) that regulate TMM in GBM. Overexpressing miR-490 causes the formation of telomere dysfunction-induced foci and DNA damage in GBM cells [58]. 

## 4. Discussion

The protein component of the telomerase complex, hTERT, is a reverse transcriptase ribonucleoprotein enzyme responsible for maintaining telomere length in cells with high replicative potential [7]. Telomeres are repetitive sequences at the ends of chromosomes that protect them from being recognized as DNA damage, acting as protective caps [7,8]. Without telomere maintenance, chromosomes shorten with each cell division, leading to cellular senescence, apoptosis, or mitotic cell death, known as “telomere crisis” [5]. 

Telomerase activation allows cells to divide indefinitely, conferring cellular immortality. Discovered in 1985, telomerase was soon identified as playing a crucial role in cancer, being expressed in about 90% of tumors but silenced in most somatic cells. These mutations are prevalent in astrocytic tumors, with significant frequency in IDH-wildtype GBMs (up to 80%) and IDH-mutant GBMs (up to 28%) [5,8,62]. hTERT promoter mutations correlate with poor prognosis in IDH-wildtype astrocytomas, leading to their reclassification alongside GBMs. 

In malignant glioma, IDH-wildtype, a major mechanism for telomerase upregulation is an activating mutation in the hTERT gene, which encodes the active catalytic subunit of telomerase. These mutations, typically at hotspots C228T or C250T, create new binding sites for GA-binding protein transcription factors (GABPs), thereby promoting hTERT mRNA transcription (see Figure 5).

Recent studies on telomerase mutations and the potential role of their inhibition have generated considerable enthusiasm among neuro-oncology researchers, leading to several experimental approaches. Nevertheless, this review highlights that a limited number of specific telomerase inhibitors are currently available, as exemplified in Table 1, Table 2, Table 3, Table 4 and Table 5. Nevertheless, this review indicates that a limited number of specific telomerase inhibitors are currently available.

Several compounds have been studied to inhibit the hTERT transcriptional level [5,15,16,17,22]. However, none of these molecules have been tested in clinical trials, despite promising results. These limitations may be due to the complexity of transcriptional hTERT regulation and the multiple key pathways that inhibit it. These pathways are not fully understood. This absence of specificity has hindered clinical progress, resulting in unmet clinical expectations.

Several authors have proposed some molecules for the direct inhibition of hTERT [26,27,28]. Unfortunately, despite promising results in vitro, some of these compounds have unfavorable pharmacokinetic properties, such as poor cellular permeability, preventing their advancement to the clinic [5]. Furthermore, the type of GBM cell lines and the model system created by each author were different, making the results extremely heterogeneous and difficult in clinical application (Appendix A includes details about the cell lines and the model system).

Additionally, these studies have some limitations, such as the lengthy period required for telomere shortening to trigger senescence and apoptosis. This allows cancer cells to activate the alternative lengthening of telomeres (ALT) mechanism, undermining attempts to inhibit telomerase. In response to the poor pharmacokinetic properties, some authors investigated an indirect approach that has been explored using the G-quadruplex DNA stabilizers, which stabilize guanine-rich secondary structures formed at telomeric repeats and thus inhibit telomerase activity [8,46,48]. However, it is imperative to conduct additional clinical studies in order to comprehend the precise mechanism of action of these compounds.

IMT is the only anti-telomerase drug to have entered clinical trials. This compound was developed using rational drug design as a competitive enzyme inhibitor. IMT is extremely potent in reducing telomerase activity in cell-free biochemical assays and in various cancer cell lines [38]. It has also been demonstrated that it inhibits the proliferation of breast, lung, and myeloma cells in vitro and in vivo, leading to telomere shortening. IMT inhibited telomerase activity in GBM cells, blocked neurosphere formation in vitro, and halted tumor growth in vivo in subcutaneous xenograft models using patient-derived tumor-initiating cells. Furthermore, IMT has also been tested in combination therapies with TMZ and RT, resulting in a reduction in tumor activity by 70% within 3–5 days [38,39] (Table 6). Despite these promising results, clinical studies of IMT have not demonstrated the same level of success in solid tumors. In 2015, IMT was designated as an orphan drug for myelofibrosis; however, it was not approved by the FDA. 

Several authors have tested the effectiveness of inhibiting hTERT, in combination with other adjuvant treatments (summarized in Table 6). Das et al. reported an improvement in sensitivity to IFN-γ and Taxol in GBM cell lines treated with retinoids [23,52]. Bejarano et al. demonstrated the antitumor effect of inhibiting TFR1 and showed enhanced antitumor effects when combined with gamma-irradiation and TMZ [43]. Furthermore, Berardinelli et al. investigated the combination of RHPS4 with carbon ion beam treatment, which showed potentiated radiation effects in terms of cell killing [46].

**Table 6 ijms-25-08700-t006:** List of studies on hTERT inhibition as a single therapy vs. combined therapies.

Author	Drug	hTERT Inhibition Alone	Combined Therapies		
			Chemo Therapies	Radio Therapies	Others Drug
Mirzazadeh et al. [14]	Resveratrol (RSV)	✔			
Gurung et al. [15]	Thymoquinone (TQ)	✔			
Khaw et al. [16]	Curcumin	✔			
Khaw et al. [17]	Plumbagin	✔			
Khaw et al. [18]	Genistein	✔			
Khaw et al. [19]	Trichostatin A (TSA)	✔			
Lin et al. [20]	Butylidenephthalide (BP)	✔			
Kiaris and Schally et al. [21]	MZ-5-156 (GH-RH antagonist)	✔			
Udroiu et al. [22]	Epigallocatechingallate (EGCG)	✔			
Das et al. [23]	Retinoids	✔			✔ (Improved IFN-γ and Taxol sensitivity)
Aquilanti et al. [5]	CRISPRi approach	✔			
Ergüven et al. [24]	Suramin	✔			
Lavanya et al. [26]	BIBR1532	✔			
Biray Avci et al. [28]	BIBR1532	✔			
Ahmad et al. [29]	Costunolide	✔			
Gurung et al. [30]	MST-312	✔			
Takahashi et al. [32]	Eribulin	✔			
Cheng et al. [33]	Arsenico	✔			
Marian et al. [38]	Imetelstat	✔	✔ (TMZ)	✔	
Ferrandon et al. [39]	Imetelstat	✔		✔	
Ozawa et al. [40]	GRN163	✔			
Hashizume et al. [41]	GRN163	✔			
Bejarano et al. [43]	TRF1 Inhibition	✔	✔ (TMZ)	✔	
Zhou et al. [44]	BRACO-19	✔			
Lagah et al. [45]	RHPS4	✔			
Berardinelli et al. [46]	RHPS4	✔		✔	
Hasegawa et al. [47]	Telomestatin	✔			
Merle et al. [48]	N-methylated triflate (TAC)	✔		✔	
Mancini et al. [55]	siRNA GABPβ1L inhibition	✔			
George et al. [52]	hTERT siRNA + IFN-γ	✔			✔ (IFN-γ)
Falchetti et al. [53,54]	hTERT siRNA	✔			
Lavanya et al. [26]	hTERT siRNA	✔			
Wang et al. [56]	Anti-miR21	✔			
Kim et al. [57]	shMUC1	✔			
Vinchure et al. [58]	miR-490	✔			

This review highlights the challenges of developing compounds that effectively inhibit hTERT and have a meaningful impact on patient care. This is substantiated by the insufficient number of clinical trials, with only two trials being identified on ClinicalTrial.gov (NCT04280848; NCT03491683).

Reardon et al. reported the utilization of telomerase to elicit antitumoral immune responses, encompassing novel targets and future perspectives on inhibiting hTERT. This approach is subject to limitations due to the well-known resistance of GBM to immune checkpoint blockade. This resistance may be due a low tumor lymphocyte infiltration and low expression of inhibitory markers such as PD-L1. Telomerase-based vaccines such as INO5401 have been evaluated in clinical trials and have shown promising results in combination with standard care [63,64].

J. Maggio et al. discovered that the loss of PIN1 in the GBM cell line (LN-229) leads to decreased malignant behavior and tumorigenicity, both in vitro and in vivo. This supports the idea that PIN1 plays a key role in the GBM model. Additionally, the study showed that the presence of PIN1 affects telomeric dynamics by downregulating hTERT expression and telomerase activity in GBM. These results provide a basis for the design and development of new therapies for GBM, with PIN1 as a novel target for the treatment of this disease [65].

Shi et al. also focused on fumarate generated by arginosuccinate lyase (ASL), which functions as an oncometabolite, enhancing hTERT transcription. This finding suggests that targeting the metabolic activity of ASL could be a potential therapeutic approach for patients with GBM [66].

Takahashi et al. examined in vitro the cytotoxic effect of Eribulin on the TERT-mutated GBM cell lines U87MG, U251MG, U118MG, and LN229. The results indicated that Eribulin suppressed the growth of all GBM cells tested in a dose-dependent manner. In this study, the authors evaluated whether Eribulin inhibited the growth of intracerebrally xenografted luciferase-expressing U87MG cells (U87MG-Fluc2) by using an in vivo imaging system. These results indicated that Eribulin inhibited the growth of U87MGFluc2 tumors in vivo [32].

## 5. Limitations

In this study, we examine the effects of telomerase inhibition in several types of GBM cell lines with high heterogenicity of GBM cell model (Appendix A). The most used GBM cell line was U87MG. Cell lines may represent a different condition from the in vivo situation of a complex, multidimensional tumor with multiple subpopulations of cells. Furthermore, they exhibit a selection bias resulting from prolonged cultivation, which may hold particular significance in relation to telomerase. Only a small proportion of the papers that were quoted in the work used fresh tumor materials or in vivo models.

## 6. Conclusions

Our study conducted a systematic review of preclinical trials that evaluated specific telomerase inhibitors for the treatment of glioblastoma. 

Although some compounds have demonstrated promising results in preclinical studies, their clinical translation has been hindered by the complex regulation of hTERT and unfavorable pharmacokinetic properties. 

At present, there are no approved telomerase inhibitors for GBM, and clinical trials have been unsuccessful. Imetelstat, which has shown potent telomerase inhibition in various cancer models, has not achieved significant success in solid tumors, including GBM. Telomerase-based vaccines and multi-target inhibitors possess the potential to provide more efficacious strategies for targeting telomerase and other pathways involved in tumor progression.

## Figures and Tables

**Figure 1 ijms-25-08700-f001:**
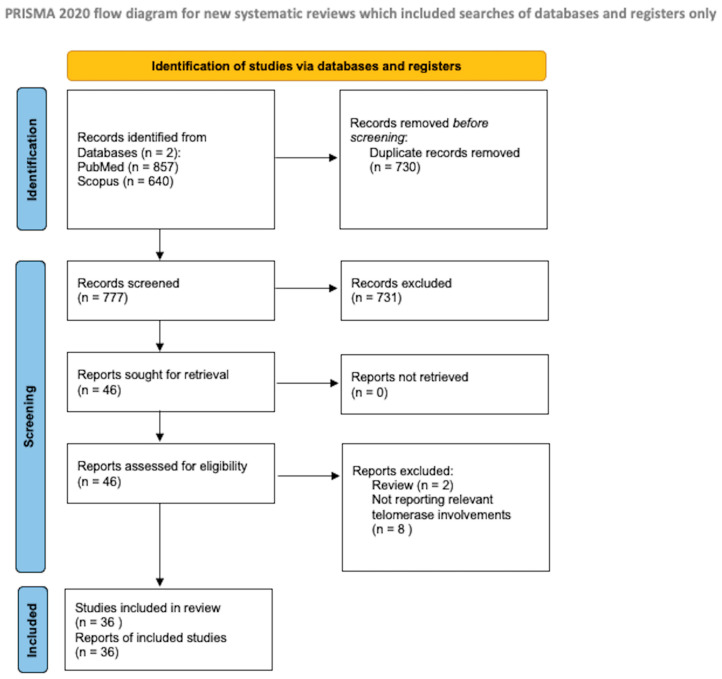
PRISMA flow diagram [11].

**Figure 2 ijms-25-08700-f002:**
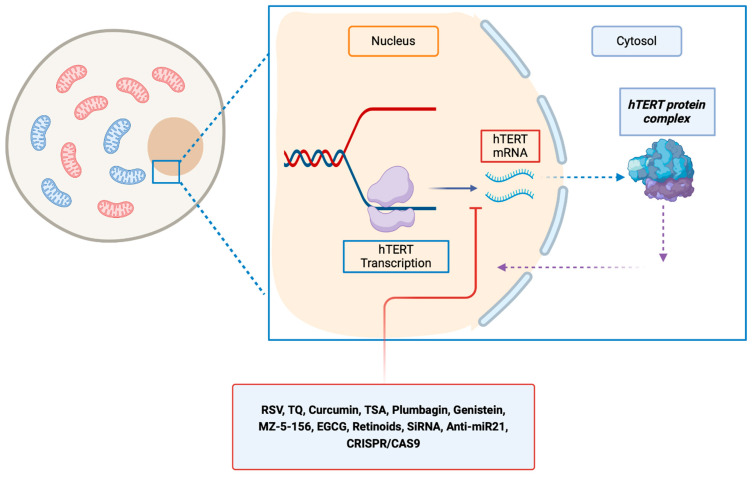
Molecules that inhibit hTERT at the transcriptional level, resulting in a reduction in hTERT mRNA levels. EGCG, epigallocatechin gallate; hTERT, human telomerase reverse transcriptase; hTER, RSV, resveratrol; TQ, thymoquinone; TSA, trichostatin A; MZ-5-156: growth hormone–releasing hormone antagonist inhibitor.

**Figure 3 ijms-25-08700-f003:**
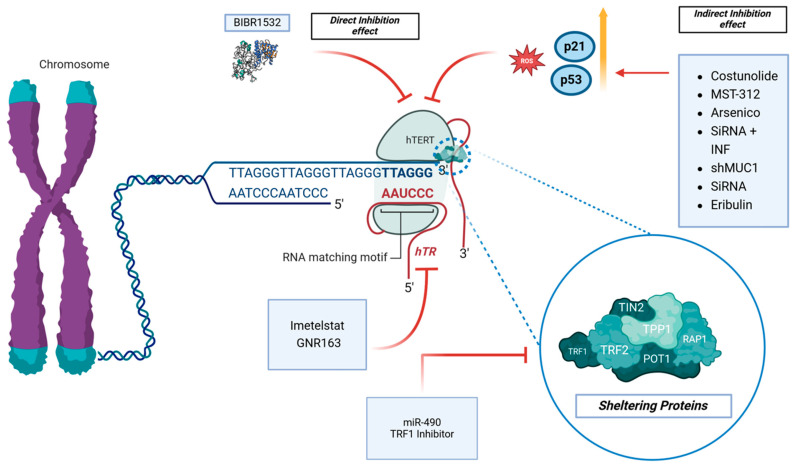
The telomerase pathway and its inhibitors are the topic of interest. BIBR1532 has direct effects on the hTERT protein, while other molecules have an indirect effect through the modulations of ROS, p53, and p21. Imetelstat and GNR163 affect hTR, while the TRF1 Inhibitor and miR-90 affect the sheltering proteins complex.

**Figure 4 ijms-25-08700-f004:**
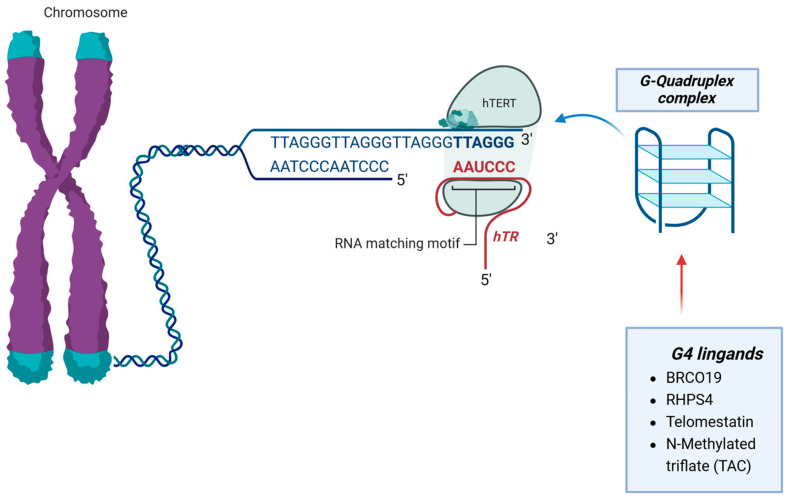
G4 ligands bind to DNA secondary structures called the G-quadruplexes complex. These proteins interact with telomeres, induce telomere uncapping, and indirectly affect telomerase function.

**Figure 5 ijms-25-08700-f005:**
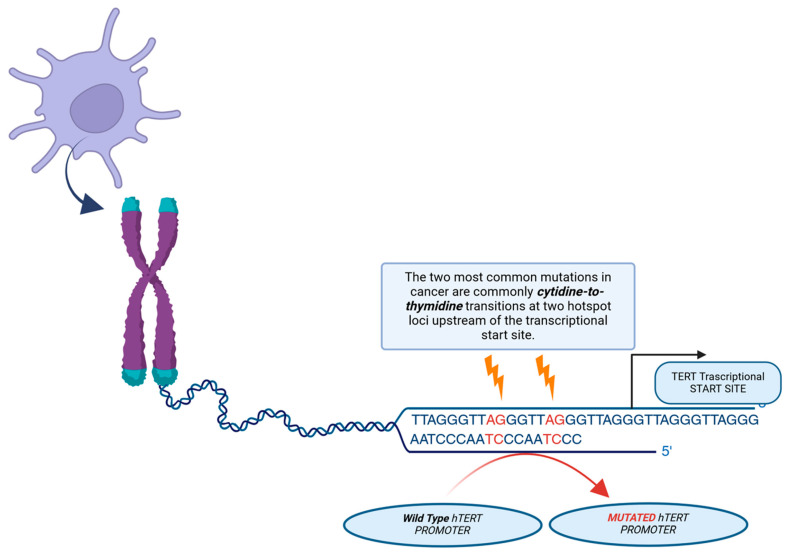
Illustration of the mechanism of action of telomerase reverse transcriptase (h*TERT*) promoter mutations. Two of the most common mutations in cancer are mutations in the cytidine-to-thymidine transition, which occur in two hotspots before the transcription start site. A novel binding site for the GABP transcription factor, which activates TERT expression, is created by these mutations.

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
