# Peer review of "Advancements in Telomerase-Targeted Therapies for Glioblastoma: A Systematic Review"

_ijms, 2024, doi:10.3390/ijms25168700_

Round 1

Reviewer 1 Report

Comments and Suggestions for Authors

The authors provide a systematic review of the current status of Telomerase-targeted therapies in Glioblastoma. The authors summarize findings from published literature in GBM, categorized by the mechanism-of-action of the drugs studied. 

Major comments:

1.        The authors could briefly mention how the studies mentioned in the review studied telomerase activity (kind of assay, etc)

2.        It would be important to mention the use of Eribulin in a clinical trial for recurrent GBM, and its results.

3.        The authors could also data table for studies that use telomerase inhibition as a single therapy vs. combined with other therapies (chemo/radiation/angiogenesis inhibtion), etc

4.        While authors outline in their tables whether in vitro or in vivo studies were conducted, it might also be useful to categorize model systems (2D vs 3d/organotypic models system; cell lines-radio/chemoresistant vs. radio/chemosensitive; syngeneic vs humanized vs scid models). It will be easier to see which systems show promising effects and whether correct systems were used for the study. 

Minor comments:

1.        Correct spelling to “cytosol” in Figure 2 schematic

2.        In lines 287-288- name the GBM cell line used in this paper

3.        Lines 464-465 are a repetition of the sentence before it.

Comments on the Quality of English Language

Minor English edits required, as outlined in my review report.

Author Response

Reviewer  1

=====================

The authors provide a systematic review of the current status of Telomerase-targeted therapies in Glioblastoma. The authors summarize findings from published literature in GBM, categorized by the mechanism of action of the drugs studied.

Major comments:

The authors could briefly mention how the studies mentioned in the review studied telomerase activity (kind of assay, etc)

  • Answer: the author improved any paragraph of the manuscript with biological details of the technique applied for single authors to analyze the telomerase activity and other relevant data.

It would be important to mention the use of Eribulin in a clinical trial for recurrent GBM, and its results.

  • Answer: The authors thank you for this comment however. The use of Eribulin is reported in section: Molecules that inhibit direct or indirect the hTERT. It is summarized in Table 2 and discussed in the text from lines 227 to 241. However, the authors, based on the suggestion received, have also added to the discussion from lines 563 to 569.

The authors could also data table for studies that use telomerase inhibition as a single therapy vs combined with other therapies (chemo/radiation/angiogenesis inhibition), etc

  • Answer: The authors are grateful for your comment and have provided Table 6 to compare studies with telomerase inhibition as either a single therapy or combined with other therapies. Furthermore, this table is outlined in the section Discussion from lines 530 to 540.

While authors outline in their tables whether in vitro or in vivo studies were conducted, it might also be useful to categorize model systems (2D vs 3d/organotypic models system; cell lines-radio/chemoresistant vs. radio/chemosensitive; syngeneic vs humanized vs scid models). It will be easier to see which systems show promising effects and whether correct systems were used for the study.

  • Answer: The authors are grateful for your comment and have provided Supplementary Material where each paper is detailed with a focus on cell lines applied and the GBM system model used.

Minor comments:

Correct spelling to “cytosol” in Figure 2 schematic

  • Answer: The authors corrected the spelling mistake and uploaded the Figure.

In lines 287-288- name the GBM cell line used in this paper

  • Answer: The authors thank you for this comment and have corrected the error. As reported before, any details about the cell lines used are scheduled in Supplementary Material.

Lines 464-465 are a repetition of the sentence before it.

  • Answer: The authors thank you for this comment and have corrected the error. As reported before, any details about the cell lines used are scheduled in Supplementary Material.

Reviewer 2 Report

Comments and Suggestions for Authors

In the paper, „ Advancements in Telomerase-Targeted Therapies for Glioblastoma: A Systematic Review “, the authors Giovanni Pennisi et al. present a very large and nice and insightful study regarding molecular oncology in glioblastoma (GBM); the work is very informative and conclusive, giving a great overview on the current status of knowledge. Many papers have been reviewed, analysed and included in a diligent way, however, some aspects should still be improved.

The authors focus on adult GBM, in children, GBM usually is a completely different disease, based on molecular biology. Here, hTERT is less relevant; this should be mentioned.

The papers are sorted by the compounds used for hTERT inhibition, not by the model they used. As the authors describe in most cases, practically all papers analysed effects of telomerase inhibition in just one or two cell lines, in particular long term in vitro cultivated cell lines like U87MG.

It should be mentioned that such cell lines might represent a different condition as compared to the in-vivo situation of a complex multi-dimensional tumor containing multiple subpopulations of cells. Further, they contain a selection bias due to long-term cultivation, which may be of particular relevance regarding telomerase.

Only a small proportion of papers that were quoted in the work used fresh tumor materials or in-vivo models. In fact, in the conclusion, the authors state that “Our study conducted a systematic review of clinical trials that evaluated specific telomerase inhibitors for the treatment of Glioblastoma”. However, as far as I can see, the work deals with preclinical work only; and in fact, the authors state that “none of these molecules have been tested in clinical trials, despite promising results.”

This should be corrected.

Minor point: The authors do not refer to the review “TERT Promoter Alterations in Glioblastoma: A Systematic Review” by Nathalie Olympios et al., Cancers (Basel). 2021 Mar; 13(5): 1147.

Comments on the Quality of English Language

good

Author Response

Reviewer  2

=====================

In the paper, „ Advancements in Telomerase-Targeted Therapies for Glioblastoma: A Systematic Review “, the authors Giovanni Pennisi et al. present a very large and nice and insightful study regarding molecular oncology in glioblastoma (GBM); the work is very informative and conclusive, giving a great overview on the current status of knowledge. Many papers have been reviewed, analyzed and included in a diligent way, however, some aspects should still be improved. The authors focus on adult GBM, in children, GBM usually is a completely different disease, based on molecular biology. Here, hTERT is less relevant; this should be mentioned.

  • Answer: The authors thank you for this suggestion and they added from lines 90 to 100 and reference n°11-12.

Only a small proportion of papers that were quoted in the work used fresh tumor materials or in-vivo models. In fact, in the conclusion, the authors state that “Our study conducted a systematic review of clinical trials that evaluated specific telomerase inhibitors for the treatment of Glioblastoma”. However, as far as I can see, the work deals with preclinical work only; and in fact, the authors state that “none of these molecules have been tested in clinical trials, despite promising results.” This should be corrected.

  • Answer: The authors consider this comment very important and have added the Limitations section where this bias deriving from the analysis of the studies in the literature is reported.

The papers are sorted by the compounds used for hTERT inhibition, not by the model they used. As the authors describe in most cases, practically all papers analysed effects of telomerase inhibition in just one or two cell lines, in particular long term in vitro cultivated cell lines like U87MG.It should be mentioned that such cell lines might represent a different condition as compared to the in-vivo situation of a complex multi-dimensional tumor containing multiple subpopulations of cells. Further, they contain a selection bias due to long-term cultivation, which may be of particular relevance regarding telomerase.

  • Answer: The authors thank you for this comment and have corrected the error in the conclusion.

Minor point: The authors do not refer to the review “TERT Promoter Alterations in Glioblastoma: A Systematic Review” by Nathalie Olympios et al., Cancers (Basel). 2021 Mar; 13(5): 1147.

  • Answer: The authors have included this interesting review in the materials and methods to explain the reasons for the exclusion of glioblastoma in pediatric patients.